# Research into New Molecular Mechanisms in Thrombotic Diseases Paves the Way for Innovative Therapeutic Approaches

**DOI:** 10.3390/ijms25052523

**Published:** 2024-02-21

**Authors:** Sara Sacchetti, Chiara Puricelli, Marco Mennuni, Valentina Zanotti, Luca Giacomini, Mara Giordano, Umberto Dianzani, Giuseppe Patti, Roberta Rolla

**Affiliations:** 1Clinical Chemistry Laboratory, “Maggiore della Carità” University Hospital, Department of Health Sciences, University of Eastern Piedmont, 28100 Novara, Italy; sara.sacchetti@uniupo.it (S.S.); valentina.zanotti@uniupo.it (V.Z.); 20023552@studenti.uniupo.it (L.G.); mara.giordano@med.uniupo.it (M.G.); umberto.dianzani@med.uniupo.it (U.D.); roberta.rolla@med.uniupo.it (R.R.); 2Division of Cardiology, “Maggiore della Carità” University Hospital, Department of Translational Medicine, University of Eastern Piedmont, 28100 Novara, Italy; marcog.mennuni@maggioreosp.novara.it (M.M.); giuseppe.patti@uniupo.it (G.P.)

**Keywords:** atherosclerotic plaque, coagulation cascade, platelets, extracellular vesicles, genetic factors, inflammation, platelet–endothelial interaction, therapies, thrombosis

## Abstract

Thrombosis is a multifaceted process involving various molecular components, including the coagulation cascade, platelet activation, platelet–endothelial interaction, anticoagulant signaling pathways, inflammatory mediators, genetic factors and the involvement of various cells such as endothelial cells, platelets and leukocytes. A comprehensive understanding of the molecular signaling pathways and cell interactions that play a role in thrombosis is essential for the development of precise therapeutic strategies for the treatment and prevention of thrombotic diseases. Ongoing research in this field is constantly uncovering new molecular players and pathways that offer opportunities for more precise interventions in the clinical setting. These molecular insights into thrombosis form the basis for the development of targeted therapeutic approaches for the treatment and prevention of thrombotic disease. The aim of this review is to provide an overview of the pathogenesis of thrombosis and to explore new therapeutic options.

## 1. Introduction

Coronary heart disease (CHD) is the third leading cause of death worldwide, responsible for approximately 17.8 million deaths annually [1,2]. Atherothrombosis is a vascular arterial disease process involving coronary, cerebral and peripheral arteries and is a key contributor to cerebral, cardiac and peripheral clinical events. Chronic inflammation and the deposition of lipids, cholesterol and inflammatory cells in the arterial walls lead to the formation of atherosclerotic plaques [3]. Rupture or erosion of these plaques exposes thrombogenic material, including tissue factor and collagen, which favors platelet adhesion and activation. The platelets adhere to the exposed subendothelial components, initiating the coagulation cascade and ultimately leading to the formation of a thrombus in the artery [3,4].

Clinical manifestations directly correlate with the affected organ system: stroke, myocardial infarction (MI) and limb ischemia for cerebral, coronary and peripheral arteries, respectively [5].

Venous thromboembolism (VTE) is a common and serious complication of hospitalization that is associated with increased morbidity and mortality. European epidemiologic studies report an overall incidence of 110 to 130 cases of VTE per 10,000 patients per year [1], with 10–12% of deaths per year attributable to VTE [6]. Venous thrombosis, which is characterized by the formation of blood clots in deep veins, involves factors known as Virchow’s triad: blood stasis, endothelial damage and hypercoagulability [7]. Conditions such as immobility and prolonged bed rest or diseases such as congestive heart failure often lead to reduced blood flow, which promotes the formation of blood clots due to stasis [7]. Endothelial injury to the veins caused by trauma, surgery, inflammation or underlying vascular disease triggers reactions that promote blood clot formation [7]. Hypercoagulability, which increases the blood’s ability to clot and is caused by genetic predispositions, malignant diseases, autoimmune diseases or the use of hormonal contraceptives, contributes to an increased risk of thrombosis [8].

Both arterial and venous thrombosis trigger an inflammatory response involving cytokines, chemokines and the recruitment of immune cells. Endothelial dysfunction or damage has been shown to be a critical component in the development and progression of both venous and arterial vascular disease.

Understanding the intricate mechanisms underlying arterial and venous thrombosis is critical as these diseases contribute significantly to global morbidity and mortality. The interplay between atherosclerosis and inflammatory responses underscores the complex nature of thrombotic events. Developing targeted strategies to address these multiple factors is critical to advancing preventive and therapeutic interventions for cardiovascular and venous disease.

## 2. Endothelial Dysfunction

The endothelium, which surrounds the arterial, venous and lymphatic vessels, is not a barrier but is actively involved in various hematological and metabolic functions. In a healthy state, the endothelium protects against blood clotting, tissue damage and ischemic disease by maintaining a quiescent state through laminar blood flow and cytoprotective factors [9]. In its role as a regulator of vascular tone, the endothelium produces various vasoactive mediators to modulate vascular smooth muscle contraction. The major vasoconstrictor factors include thromboxane A2 and endothelin-1, while the major vasodilator factors are nitric oxide (NO), prostacyclin and endothelium-derived hyperpolarizing factor [9]. Under normal conditions, the endothelium prevents thrombosis through a number of anticoagulant and antiplatelet mechanisms. To prevent excessive clotting, endothelial cells produce regulatory surface proteins, including thrombomodulin (TM), endothelial protein C receptor (EPCR), tissue factor pathway inhibitor (TFPI) and protein C (PC) [9]. When TM is bound to thrombin, it converts the PC bound to the EPCR into activated protein C (APC). APC acts together with protein S (PS) to inactivate activated FVIII and FV. This restricts the functions of the intrinsic tenase complex (FVIIIa-FIXa) and the prothrombinase complex (FXa-FVa). In addition, TFPI plays a role in inhibiting the tissue factor (TF)-FVIIa activation of FX. Endothelial cells also play a crucial role in the regulation of blood coagulation by synthesizing, storing and secreting multimeric strands of von Willebrand factor (VWF) and coagulation factor VIII (FVIII). In summary, endothelial cells make an important contribution to the regulation of blood coagulation by producing important regulatory proteins that control various steps of the coagulation cascade, thereby maintaining hemostatic balance and preventing excessive clot formation [9]. However, impairments such as biochemical and physical factors can affect endothelial function, leading to a condition known as endothelial dysfunction [10]. Endothelial dysfunction is manifested by five key mechanisms: loss of vascular integrity, increased expression of adhesion molecules, a pro-thrombotic phenotype, cytokine production and upregulation of human leukocyte antigen molecules (HLA). This is a spectrum of phenotypes associated with different types and degrees of injury. Recent RNA sequencing studies have identified several subtypes of endothelial cells in diseases such as aneurysms and atherosclerosis. In addition, endothelial dysfunction leads to alterations in circulating endothelial colony-forming cells and endothelium-derived microvesicles [10]. Endothelial dysfunction is also defined as the loss of production of vasodilator and vasoconstrictor molecules (NO and endothelin-1) [11].

In recent decades, advancements in identifying the genes and pathways associated with endothelial function have expanded our understanding of preventing thrombotic states [11]. Within the mitochondria, the thioredoxin system functions to inhibit the production of reactive oxygen species (ROS). In a murine model, the loss of thioredoxin reductase 2 (Txnrd2) results in a prothrombotic endothelium, leading to the development of systemic microthrombi in mice lacking this protein [12].

Another study highlights the role of ATG7 (autophagy-related 7) in endothelial cells, revealing that its deletion attenuates thrombosis and reduces the expression of tissue factor. Autophagy emerges as a potential avenue for reducing thrombotic events [13]. The Siert1/FoxO1 pathway plays a crucial role in regulating endothelial cell autophagy, increasing the release of VWF with a pro-thrombotic function. This pathway could serve as a promising target for preventing a thrombotic state [14]. Additionally, miR-181b and Card 10 (caspase recruitment domain family member 10) appear to contribute to the regulation of thrombin-induced endothelial cell activation and arterial thrombosis. The overexpression of miR-181b inhibits thrombin-induced activation of NF-kB signaling by targeting Card10 [15]. These findings underscore the intricate molecular mechanisms involved in endothelial dysfunction and its impact on the thrombotic process. Activation of the endothelium leads to a pro-thrombotic phenotype through the upregulation of coagulation factors (Figure 1). As a result, there is an increase in tissue factor on the cell surface, initiating the extrinsic pathway of coagulation. There is also an increase in inflammatory mediators such as TNF-alpha, CD40 ligand and other cytokines [16]. Activated endothelium also induces platelet mobilization through the release of VWF, P-selectin and angiopoietin-2. It has been demonstrated that activated endothelium-derived VWF has pro-thrombotic activity, while the contribution of platelet-produced VWF is comparatively lower [16,17]. This underlines the complicated interplay between endothelial dysfunction, inflammation and thrombotic processes.

The role of NO in the well-being of the endothelium is well known. A healthy endothelium produces NO, acting as a local antiplatelet factor by limiting platelet adhesion and activation. The substantial expression of eNOS by platelets suggests that both autocrine and paracrine signaling occur within developing clots, thereby limiting their progression. Consequently, drugs enhancing NO may offer cardiovascular and antiplatelet benefits, with ongoing evolution in this research area [18,19].

Recently, it has been investigated how the gut microbiota can affect the endothelium, leading to a dysfunctional state [20,21]. By stimulating the enteric nervous system, bacteria can impact brain centers regulating the cardiovascular system, and through the bloodstream they modulate the circulatory system’s homeostasis. While short-chain fatty acids (SCFAs) exhibit a positive effect on the endothelium, some metabolites, such as trimethylamine (TMA), may be potentially harmful. Its oxidized form, trimethylamine N-oxide (TMAO), possesses pro-atherogenic properties and serves as a biomarker predicting cardiac risk, stroke risk and death. Endothelial dysfunction and platelet activation are implicated through the NF-kB pathway in reaching this perilous state. Furthermore, TMAO can downregulate the expression of IL-10, a well-known anti-inflammatory cytokine. TMAO also increases ROS generation and reduces NO. Additionally, TMAO compromises the endothelial cells’ self-healing ability, leading to an irreversible dysfunctional state [22,23]. When NO synthesis is inhibited, the endothelium can be restored to a healthy state through treatment with polyphenols, which can restore NO production and consequently reduce ROS [24].

Comprehensive research into the multifaceted role of endothelial dysfunction in thrombotic diseases offers valuable perspectives for researchers and clinicians, along with new therapeutic strategies.

## 3. Platelets and Endothelial Inflammation

Platelets, traditionally known for their pivotal role in hemostasis and coagulation, go beyond these functions and actively participate in various physiological processes. They contribute to both innate and adaptive immunity and play a role in neurogenesis by releasing bioactive molecules from their granules. However, the multifaceted functions of platelets also mean that platelet dysfunction can contribute to various diseases [25]. Platelets are involved in thrombosis and play an important role in mediating diseases such as MI, stroke, VTE, vascular disease, atherothrombosis, metabolic syndrome, cancer, autoimmune diseases and neurodegenerative diseases [26]. 

Atherosclerosis is an inflammatory disease of the vessel wall, whereby the immune system plays a decisive role in both the development and progression of atherosclerotic plaques. Platelets, commonly regarded as “inflammatory cells”, play a substantial role in the development of inflammatory diseases. The link between inflammation and thrombosis is evident, as inflammation can trigger thrombosis, and vice versa, giving rise to the concept of thrombo-inflammation. 

The link between platelet activation and atherothrombosis is generally recognized. When plaques rupture or erode, exposing subendothelial type I collagen, platelet activation occurs. Subsequent platelet aggregation leads to the formation of a thrombus, which is rich in platelets and contributes to ischemic disease [27]. Traditionally, this was thought to represent the extent of platelet involvement in acute cardiovascular disease [27]. As a result, antiplatelet therapy has become a standard of care for individuals at high risk of atherothrombosis [27].

However, platelets not only play a role in the subsequent thrombotic complications of atherosclerosis but also in the initiation and spread of atherosclerosis. 

Platelets actively participate in the vascular immune response and play a crucial and primary role. They are initially triggered by vascular damage and are the first to release cytokines from the alpha granules [25]. After activation, platelets express various receptors, including P-selectin, integrins such as glycoprotein (GP) IIb/IIIa and Toll-like receptors, which trigger immunological activities. Platelets have the ability to activate leukocytes, endothelial cells and smooth muscle cells (SMCs) via various receptors and secrete mediators such as adenosine diphosphate (ADP), thromboxane A2 (TXA2), cytokines such as IL-1b and TGF-β and chemokines such as CCL5 and CXCL12 [4]. When platelets adhere to activated endothelial cells, this triggers further activation in the endothelial cells. Both platelets and endothelial cells express or release chemokines. Activated endothelial cells exhibit surface markers such as ICAM-1, VCAM-1, E-selectin and P-selectin and also release chemokines such as MCP-1 and interleukin (IL)-8. In addition, both activated platelets and endothelial cells actively secrete IL-1, a potent inflammatory cytokine, and CD40L [28].

The adhesion of platelets to the endothelial surface generates signals that attract monocytes to the site of inflammation. The interaction between monocytes and platelets via P-selectin and P-selectin glycoprotein ligand-1 (PSGL-1) leads to the secretion of chemokines, cytokines, tissue factor and proteases by the monocytes. This process induces the transformation of monocytes into macrophages, which is an advanced step in the development of a mature plaque. RANTES, one of the participants in this atherogenic mechanism, is produced by activated platelets. It is an important platelet-derived chemotactic agent and promotes leukocyte activation and P-selectin-mediated recruitment of monocytes to the inflamed endothelium [28]. The cytokine PF4 is another common component of platelet alpha-granules and is rapidly released after platelet activation. PF4 stimulates the adhesion of monocytes to activated endothelial cells, promoting the differentiation of monocytes into macrophages, a crucial event in the development of atherosclerotic plaques. Moreover, platelets store large amounts of CD40L and release it upon activation [25]. Elevated serum levels of CD40L indicate an acute risk of a coronary event [28].

CD40L is an important platelet-derived cytokine and plays a crucial role in regulating the function of leukocytes. CD40L induces the expression of various inflammatory factors such as tumor necrosis factor-alpha (TNF-α), interferon-gamma (IFN-γ), IL-1, IL-6, IL-8, MCP-1, RANTES and isoforms of macrophage inflammatory protein (MIP) in various cell types, including peripheral blood monocytes [25]. In addition, CD40L contributes to enzymatic processes that reduce the stability of atherosclerotic plaques by promoting the production of matrix metalloproteinases (MMPs) by macrophages, leading to the degradation of protective fibrous caps. Moreover, platelet-derived CD40L influences thrombosis by promoting the production of TF by monocytes and macrophages. The release of platelet-derived CD40L leads to inflammatory reactions in the endothelium. The binding of CD40L to endothelial CD40 leads to the release of IL-8 and MCP-1, important chemo attractants for neutrophils and monocytes. CD40 activation on endothelial cells increases the expression of endothelial adhesion receptors such as E-selectin, VCAM-1 and ICAM-1, facilitating firm adhesion of neutrophils, monocytes and lymphocytes [28].

In addition, activated platelets bind to circulating leukocytes in the bloodstream. The adhesion of leukocytes to the blood vessel wall is facilitated by platelets via the interaction between platelet P-selectin and leukocyte PSGL-1. The binding of P-selectin to PSGL-1 triggers outside-in signaling, which triggers the production of cytokines, chemokines, tissue factor, matrix metalloproteinases (MMPs) and reactive oxygen species. These outside-in signals triggered by P-selectin are transmitted via tyrosine kinases and lead to an upregulation of the expression and conformational activation of integrins, especially Mac-1, on leukocytes, thereby controlling their adhesion. Mac-1 on leukocytes and GP Ib on platelets appear to be dominant in the regulation of platelet and leukocyte adhesion in vascular injury [28]. This specificity makes them potential targets for precise interventions. 

Therefore, platelets play a central role in the recruitment of leukocytes to the vessel wall, where they adhere to activated endothelium or extracellular matrix proteins. Thrombocytes interact with their receptor (PSGL-1) on polymorphonuclear leukocytes via platelet CD62P and lead to the formation of neutrophil extracellular traps (NET), a crucial factor in thrombus formation [25].

Indeed, the DNA found in NETs is already a direct trigger of the coagulation cascade, and NET proteases can inactivate natural anticoagulant factors. Furthermore, intravascular NETs can firmly adhere to the vessel wall, opposing resistance to the blood flow and acting as a scaffold that promotes the interaction between erythrocytes, platelets and clotting factors [29].

The crucial role of oxidized low-density lipoprotein (LDL) particles in the formation of atherosclerotic plaques is well known. These particles are taken up by macrophages in the subendothelium, leading to the formation of foam cells. This triggers a cascade of pro-inflammatory intracellular signaling, which is a key event in the atherosclerotic process. Oxidized LDLs have the ability to activate circulating platelets through the activation of tissue factors. This activation leads to the formation of ROS in the platelets and facilitates the oxidative conversion of LDL into lipid peroxides [30]. The changes in the platelet lipidome, particularly in relation to membrane phospholipids (PLs), play an important role in atherosclerosis. Oxidized PLs contribute to inflammation, endothelial dysfunction, differentiation of monocytes and macrophages, plaque formation and ischemic conditions. These events activate pro-inflammatory genes and thrombosis through receptors such as the scavenger receptor CD36, platelet-activating factor receptor (PAFR) and tissue factor TFPI [30]. In patients with cardiovascular disease, LDL binding increases platelet receptors, such as CD36, leading to ROS formation and platelet activation. LDL-induced platelet activation reciprocally induces apoptosis and inflammatory responses in monocytes via the CXCL12-CXCR4-CXCR7 axis, contributing to atherogenesis [30]. Chemokines such as CXCL12 secreted by platelets influence inflammation and activate autocrine and paracrine responses, further contributing to thrombosis. Platelet lipids, such as triacylglycerols and ceramides, are induced and increased by the CXCL12/CXCR4-CXCR7 axis and thus contribute to CVD [30]. The increased expression of CXCR7 and CXCL12 in patients with CVD is associated with recovery and a favorable prognosis. Consequently, targeting these receptors on platelets may represent a novel strategy to treat CVD and improve outcomes [30].

In summary, the immune functions of platelets are emerging as promising targets, not only for the treatment of cardiovascular disease but also for its prevention. This growing understanding of the broader contributions of platelets to atherosclerosis, beginning with the early stages of the atherosclerotic process, challenges the conventional view and underscores the need for further investigation in this dynamic area (Figure 2).

## 4. Role of Coagulation Cascade and Anticoagulant System

The blood coagulation cascade is a finely balanced system that controls hemostasis in cooperation with cellular components such as platelets, microvesicles and other blood cells [31].

Ensuring an appropriate concentration and functionality of coagulation proteins is crucial to avoid excessive clotting and the associated risk of thrombosis. This goal is achieved by several mechanisms, including the regulation of coagulation activity and the presence of several anticoagulant mechanisms: antithrombin (AT), the PC and PS system and TFPI [32]. Thrombosis is a pathological phenomenon that impedes blood flow in the vessels and is closely associated with the development of both arterial and venous thrombosis, in which TF plays a crucial role [33]. Atherosclerotic plaques contain significant amounts of TF, typically associated with monocytes/foam cells and smooth muscle cells [34]. TF antigen may also be present in the acellular core of atheromas, most likely derived from necrotic cells. Plaque TF is functional and capable of binding FVIIa, and in atherosclerosis the blood is separated from TF only by a thin monolayer of endothelial cells. MI is triggered by the rupture of an atherosclerotic plaque in a coronary artery, exposing TF to FVII/FVIIa in the blood. Inflammatory mediator-induced expression of TF in blood vessels may play a crucial role in thrombotic diseases [33]. TF expression can also be increased in neoplastic cells, which can lead to cancer-associated thrombosis (Trousseau syndrome) [35].

According to epidemiologic studies, a risk factor for thrombotic disease may be elevated plasma FVII. Increased plasma FVII clotting activity (FVII:C) or increased levels of circulating FVIIa have also been described in angina pectoris, transient ischemic attacks, diabetes, uremia and peripheral vascular disease [34].

To date, there is solid evidence that mutations in genes related to the coagulation pathway contribute significantly to an increased risk of venous thrombosis. These genes include FV and FII as well as FIX (coagulation factor IX), FGG (fibrinogen), VWF and THB1 (thrombomodulin) as well as genes related to the contact phase of blood clotting (KNG1, FXI) [36].

Most of the genetic factors that contribute to thrombosis are directly related to the proteases involved in the coagulation cascade. They result from mutations in their structural genes or from changes in other genes that affect the functionality of these proteases [37].

Recent research on blood coagulation has led to the formulation of a cell-based model in which activation of blood coagulation in vivo is initiated by the exposure of cells expressing TF on their surface. This model emphasizes that the intrinsic pathway has no clear physiological role in hemostasis, redefines the function of factor XII (FXII) and highlights its involvement in pathological thrombosis [38]. While the contact activation pathway is shown to be crucial in clot formation in vitro, its role in hemostasis in vivo is remarkably minor, as shown by the absence of bleeding tendency in mice and humans lacking FXII [39]. Instead, one of the in vivo functions of the contact pathway appears to be the generation of bradykinin, an important inflammatory mediator that is produced when kallikrein cleaves high molecular weight kininogen (HMWK). Although the contact pathway is dispensable for normal hemostasis, recent evidence points to its contribution to thrombotic disorders.

FXII deficiency has been shown in animal models to protect against thrombus formation in both arteries and veins. Increased FXII, FXI or kallikrein activity in plasma is associated with atherosclerosis and MI. In animal thrombosis models, FXII deficiency reduces arterial thrombus formation and protects animals from ischemic brain damage. Humans with severe FXI deficiency have a lower risk of stroke [34].

The relationship between the structural characteristics of fibrin clots and clinical outcomes highlights the central role of the fibrin network structure in hemostasis and thrombosis. Dense networks of thin, compact fibrin fibers correlate with an increased risk of thrombosis [40], whereas coarser networks of thick fibers are associated with an increased risk of bleeding and greater susceptibility to fibrinolysis [40,41]. This understanding forms the basis for evaluating the abnormal features of fibrin clots as potential indicators of arterial and VTE risk and provides valuable insights for clinical evaluation [42,43].

Changes in the structure of the fibrin clots are associated with the disease, particularly in advanced coronary artery disease (CAD). Young men (<45 years) with early CHD have 30% less permeable, dense fibrin networks. Mild alterations in fibrin structure were observed in healthy first-degree relatives of individuals with early CHD, indicating inherited modifications in fibrin networks. Similar changes were also demonstrated in young survivors of MI, revealing increased stiffness, shorter fibers, and impaired fibrinolysis.

These findings also apply to aged individuals with advanced CHD, who show increased fiber density and resistance to fibrinolysis compared to control subjects [40]. Reduced clot permeability, lysis time (LT) and opacity, indicative of densely packed fibrin networks, are associated with CHD complications, including stent thrombosis and the no-reflow phenomenon [44]. Patients who experience no-reflow after MI also have reduced fibrin clot permeability and susceptibility to lysis [40]. Abnormal properties of fibrin clots observed during the acute phase of MI have also been documented in acute ischemic stroke, suggesting that decreased permeability and hypofibrinolysis contribute to stroke thrombosis. Importantly, Rooth et al. show that the changes observed in ischemic stroke persist beyond 60 days, suggesting persistent impairment of fibrinolysis [45].

In VTE, reduced fibrinolytic potential after the first deep vein thrombosis (DVT) predicts recurrent VTE, including pulmonary embolism (PE). Idiopathic VTE patients and asymptomatic first-degree relatives have lower clot permeability, less compaction and prolonged LT compared to controls, with more pronounced changes in patients. In particular, the fibrin clots of PE patients are more permeable, more susceptible to lysis and less compact than those of DVT patients [46].

In addition, the clot-stabilizing effect of FXIII, which is present in both cells and plasma, is thought to play a role in pathological thrombosis. Despite being a rare autosomal recessive bleeding disorder, congenital FXIII deficiency is linked to arterial and venous thrombosis [47]. The Val34Leu polymorphism in FXIII, associated with accelerated activation and fibrin cross-linking, is significantly linked to CHD risk [40]. Meta-analysis confirms its association with CHD, especially MI [48]. In coronary artery bypass grafting patients, Val34Leu correlates with reduced clot permeability and lysis efficiency [49]. The FXIII-B His95Arg variant slightly increases venous thrombosis risk [40]. FXIII-A Tyr204Phe shows elevated risk in Dutch women for MI and ischemic stroke but lacks replication in a Brazilian population [50,51]. Mezei et al. found higher FXIII activity and FXIII-A2B2 antigen levels in women with VTE, with lower FXIII-B in men with VTE, indicating gender differences [52].

Maintaining the balance between coagulation and fibrinolysis is crucial to prevent uncontrolled thrombosis or abnormal bleeding and to avert vascular occlusion and tissue damage in conditions such as acute MI, ischemic stroke and acute peripheral vascular occlusion. Endogenous fibrinolysis is a crucial physiological countermeasure against arterial or venous thrombosis, and impaired fibrinolysis contributes to acute thrombosis and the progression of chronic thromboembolic disease. In acute coronary syndrome (ACS), impaired endogenous fibrinolysis is associated with an increased risk of major adverse cardiovascular events (MACE) and cardiovascular death at 1 year [40]. In a study of 300 ACS patients, approximately 20% had impaired endogenous fibrinolysis (defined as lysis time < 3000 s by a global thrombosis test) [53]. Another study of 496 patients with ST-segment elevation myocardial infarction (STEMI) found that impaired endogenous fibrinolysis on arrival at the cath lab was a strong predictor of MI, MACE and cardiovascular death at 1 year [54]. In particular, impaired endogenous fibrinolysis (LT < 3000 s) was significantly associated with MACE, non-fatal MI, stroke and peripheral thrombosis. Studies on fibrinolytic potential in venous thrombosis indicated a twofold increase in the prevalence of thrombotic events with prolonged clot lysis time [55]. In addition, a study of 704 patients with unprovoked VTE found a correlation between clot dissolution time and the risk of recurrence of VTE in women [56].

The anticoagulant system is another mechanism that is crucial for maintaining hemostatic balance and preventing excessive blood clotting. It comprises various components, including circulating serine protease inhibitors, the PC and PS system and TFPI. AT inhibits thrombin, while PC, which is activated by TM and EPCR, inactivates factors VIIIa and Va with PS. TFPI with PS inhibits the TF/FVIIa complex [32,57,58]. Inherited or acquired hypercoagulable disorders include deficiencies in AT, PC and PS as well as resistance to APC. In particular, resistance to APC, which is specifically associated with factor V Leiden, is a predisposing factor for recurrent VTE [59,60,61,62].

Inherited or acquired PC deficiency is associated with a hypercoagulable state leading to VTE and often contributes to familial thrombophilia [60]. In central retinal artery occlusion (CRAO), deficiency of PC and PS is common, suggesting that they play a role in thrombophilia risk [63]. The increased clotting activity observed in thrombophilic patients with a familial deficiency of PC or PS is attributed to the impaired activation of PC, which is mediated by TM and PS. This dysfunction leads to increased formation of plasmatic thrombin [60]. Brouns et al. recently observed that whole blood perfusion via collagen, collagen-like peptides and fibrin surfaces with low or high GPVI dependence showed unexpected results in patients with PC deficiency and, to a lesser extent, PS deficiency compared with controls. The observed impairment included changes in platelet activation, thrombus phenotype and fibrin formation, while platelet adhesion remained unchanged. The defect was manifested by reduced phosphatidylserine exposure, impaired thrombus contraction and delayed/suppressed fibrin formation. Of note, the mechanism was independent from TM and may involve negative platelet priming by specific plasma components. Paradoxically, PC positively influences platelet and coagulation activation, which affects arterial thrombus formation [64].

In addition, the hereditary deficiency of protein C/S can trigger acute or subacute severe cerebral venous thrombosis (CVT) and is associated with increased inflammation levels, which has a negative impact on the prognosis of severe CVT. The implementation of anti-inflammatory therapy, including the administration of methylprednisolone, in conjunction with standard anticoagulant treatment, is emerging as a promising and potentially effective approach for the treatment of severe CVT in individuals with hereditary protein C/S deficiency to improve clinical outcomes without significant adverse effects [65].

Another crucial component of the anticoagulant system is PS, a plasma glycoprotein that contains γ-carboxyglutamate. It is mainly synthesized and secreted by hepatocytes, endothelial cells and Leydig cells. Approximately 2.5% of circulating PS is found in the α-granules of platelets, and platelet PS is exclusively produced by its expression in megakaryocytes [58].

PS is traditionally known for its role as a cofactor for APC and TFPI. However, a recent report sheds light on a previously overlooked function of PS, direct inhibition of factor IXa (FIXa) [66]. In addition, PS exhibits anti-inflammatory functions, particularly when it circulates in plasma and forms a non-covalent high-affinity complex with C4BP [61]. Interestingly, only the free fraction of plasma PS exhibits full anticoagulant properties, as C4BP blocks the APC cofactor activity of PS [67]. Moreover, PS serves as one of the two activating ligands for the TAM family of receptor tyrosine kinases TYRO3, AXL and MER. The other TAM ligand is GAS6, which is structurally related but does not act as an anticoagulant and is present in very low concentrations in the blood. The kinase activity and signaling of MER are activated by the binding of PS (and GAS6) to the extracellular domain of MER. Activation of MER in macrophages and other immune cells has a broad immunosuppressive effect and attenuates the production of type I interferons, IL-6, TNF and other cytokines. Deficits in TAM receptor signaling resulting from decreased receptor or PS expression are associated with chronic immune hyperactivation and contribute to hypercoagulability [68,69].

Inherited or acquired PS deficiency is considered a predisposing factor for VTE, as APC and TFPIα function less effectively without corresponding cofactors [61,66]. TFPI deficiency also increases the risk of VTE and contributes to increased mortality in cancer patients [70]. Genetic variants of TFPI can increase the risk of VTE [71].

## 5. Genetic Studies in Thrombosis

The genetic component of the risk of VTE has been known since the 1960s, with an estimated heritability of 40–60%. A minority of patients carry a variant in a limited number of genes that lead to inherited forms of thrombophilia (Table 1). The classic, well-established genetic susceptibility factors for VTE are the variants in SERPINC1, PROC and PROS, which encode the natural anticoagulants AT, PC and PS, respectively. Several private rare mutations (i.e., present in <0.001 of the population) have been discovered in these genes, which lead to deficiencies in the corresponding proteins and thus increase the risk of VTE in heterozygous carriers by up to 10 times. Conversely, factor V, factor II, ABO and FGG contribute with relatively frequent susceptibility SNPs. Factor V Leiden (Q506 in F5), with a frequency of approximately 5% in Caucasians, is associated with an approximately 3-fold increased risk in heterozygous carriers, while homozygosity (which is rare) increases the risk by 20-fold [72]. A slightly lower risk is associated with the prothrombin variant G20210A (rs1799963) in the 3′UTR of the F2 gene, which correlates with elevated prothrombin levels. This variant is present in 2% of the Caucasian population, and heterozygotes have a 2- to 3-fold increased risk of venous thrombosis. The rs1799963 has also been associated with an increased risk of arterial ischemic stroke in adults with an OR of 1.48 [73]. Although homozygosity for rs1799963 is less common than homozygosity for FV Leiden, the risk of VTE is high and has been reported to be 30-fold [59].

The ABO risk alleles are more common (approximately 30%). Blood groups A1 and B are associated with an increased risk of VTE, even if the associated risk is lower and lies between 1.5 and 2 [74]. The FGG rs2066865, located in the 3′UTR of the gene, is present in approximately 25% of the general population and is associated with an increased OR of approximately 1.4. 

In addition, two variants of FIX, FIX Padua (p. Arg384Leu) and Shanghai (p. Arg384Gln), which affect the Arg384 residue, have been associated with thrombosis. These variants lead to an increase in FIX activity, which has been associated with thrombosis. A novel FIX gene missense mutation, c.1018G > A, leading to a substitution of p. Glu340Lys (termed FIX Shanghai II), was identified in a young male patient with venous thrombosis [75].

All these genetic factors have been identified thanks to the knowledge of the role of proteins involved in coagulation pathways. Currently, the assessment of VTE risk is complex and based on a multitude of parameters that are detailed in ESC Guidelines (https://doi.org/10.1093/eurheartj/ehz405, accessed on 21 January 2023). These include the patient’s clinical history, a recent stroke, fracture and age, as well as the presence of the classic genetic polymorphisms, which are only present in a small proportion of individuals. 

As a result of technological advances in high-throughput genomics, the search for prothrombotic genetic risk factors has shifted from a candidate gene approach to genome-wide association studies (GWAS), which allow the identification of numerous common and rare genetic variants that confer susceptibility to these diseases [76,77].

Recently, two meta-analyzed GWAS [78,79] identified several new loci associated with VTE and revealed new pathways involved in thrombosis. This opens new perspectives for the development of improved antithrombotic treatments and the development of a polygenic risk score (PRS), which is expected to lead to the identification of a larger number of individuals with a genetic predisposition to thrombosis than previously thought.

Thibord et al. conducted a meta-analysis of 55,330 participants with VTE and identified 10,493 variants that reached genome-wide significance, corresponding to 68 loci that were successfully replicated. Apart from the classical gene associations and previously identified pathways, the authors discovered new genes that were not previously implicated in the pathophysiology of VTE. Some genes at these loci are involved in the plasma levels of hemostatic factor and hematologic traits (i.e., *XXYLT1*, *FUT2*, *RCOR1*, *REST*), suggesting that these proteins may interfere with the coagulation cascade [78].

Ghouse et al. performed the largest GWAS meta-analysis on VTE by combining genetic data from 81,190 cases and 1,419,671 controls and identified 93 VTE risk loci, 62 of which were novel. In addition to genes directly or indirectly involved in the coagulation cascade, the authors identified loci that play a role in platelet function and formation. Among the most important is rs10993706 in SYK, a gene encoding a tyrosine kinase involved in the activation of glycoprotein VI, which is expressed on platelets and is important for platelet activation and adhesion. Another novel finding was the rs12097293 within MTOR, which regulates platelet spreading on fibrinogen, and the rs1805081 in *NPC1*, a gene encoding the Niemann–Pick intracellular cholesterol transporter 1, which mediates cholesterol transport and is mutated in lysosomal storage disease [79].

Although most of the loci showed only modest effects, by combining the effects of all variants into a polygenic risk score (PRS) it is possible to identify both high- and low-risk individuals. By using the PRS developed by Ghouse et al., a higher proportion of individuals at significant risk was identified, comparable to that of individuals carrying monogenic thrombophilia variants. The test for FV and prothrombin mutations allows the identification of carriers with an approximately 3-fold increased risk of VTE, with only 1 in 10,000 individuals being affected. In comparison, 1 in 100 people have a 3.7-fold increased polygenic risk of VTE, so 100 times as many people with a similar risk can be identified. Although high PRS due to inherited genomic variants cannot be modified, individuals with high PRS may benefit from ongoing monitoring and lifestyle modification (i.e., non-smoking, exercise, healthy diet) [79].

## 6. Role of Extracellular Vesicles in Thrombosis

Recent technological advances, particularly in electron microscopy, have supported growing interest in extracellular vesicles (EVs), small cell-derived particles whose role extends beyond intercellular communication. These lipid bilayer structures play a critical role in physiological and pathological processes, including immune response, cancer metastasis [80], endothelial dysfunction and thrombosis [81]. Extracellular vesicles, which are classified according to their size, density and centrifugation speed [82,83,84], are divided into three main types. Exosomes (50–150 nm) form by inward budding of cell membranes and mature into multivesicular bodies (MVBs). They can be incorporated into lysosomes or fuse with the cell membrane, releasing their contents into the extracellular space. They are crucial for intercellular communication [85]. Microvesicles (100–1000 nm) detach from the plasma membrane, a process that requires the support of the cytoskeleton and molecular motors (kinesins and myosins) [85,86]. Apoptotic bodies (up to 5000 nm) are released from dying cells by mechanisms that include increased hydrostatic pressure and cellular contractions. They signal ongoing apoptosis and transfer cargoes, including DNA or factors that promote target cell survival and differentiation [87,88]. The release of EVs is not restricted to specific cell types, and surface antigens reflect their cellular origin, facilitating detection by flow cytometry. The surface markers for different EVs according to their origin are listed in Table 2 [81,89,90]. Platelet-derived EVs are the most common, followed by those from endothelial cells, granulocytes and erythrocytes [91,92].

In general, an increase in EV levels has been reported in association with various pathological conditions, along with a disease-related change in their phenotype, to such an extent that they could be considered potential biomarkers for the underlying disease [89,93,94]. Specifically, in pathological conditions characterized by a pro-thrombotic tendency, elevated levels of circulating EVs are definitely a hallmark of endothelial injury or at least dysfunction, but it seems that they also promote the progression of cardiovascular disease [90]. Indeed, elevated EV levels have been described in patients with cardiovascular risk factors such as diabetes mellitus [93] or metabolic syndrome [94], as well as in patients with established cardiovascular disease such as stroke [95] or acute ACS [96]. 

EVs may have a prothrombotic tendency per se by expressing TF [97] and negatively charged phospholipids (such as phosphatidylserine) on their surface [98,99]. TF is normally stored in caveolae, microdomains of the cell membrane characterized by a specific lipid composition and involved in the process of EV release. Phosphatidylserine interacts directly with its receptor, TIM4, which is expressed on erythrocytes and endothelial cells. Furthermore, in the presence of calcium ions, phosphatidylserine associates with FXII and promotes the formation of the tenase (FVIIIa, FIXa and FX) complex that activates FX and of the prothrombinase complex (FVa, FXa and thrombin), triggering the conversion of prothrombin into active thrombin, which eventually acts on fibrinogen to initiate the formation of a fibrin clot [98]. A procoagulant behavior is also exhibited by PSGL-1 or CD162, hidden in lipid rafts and interacting with CD62P, which is upregulated on activated platelets and endothelial cells [99].

Platelet-derived EVs are not only the most abundant circulating EVs, but also the major players in the coagulation process, with pro-coagulant activity that may even exceed that of the mother cell [100]. However, erythrocytes, especially the damaged and senescent ones [101], and immune cells such as neutrophils and monocytes can also secrete prothrombotic EVs, suggesting that the coagulation cascade is much more complex than it seems. In fact, there is a clear link between inflammation and the formation of EVs from different cell types. Proinflammatory stimuli such as TNF-α, bacterial lipopolysaccharide (LPS) or ROS are all potent triggers for the release of EVs and increase the expression of TF, which activates the coagulation cascade in a self-reinforcing cross-talk [102]. 

The function of EVs as signaling mediators also leads to the intercellular transfer of key molecules involved in the coagulation process and contributes to bridging the link between inflammation and thrombosis, recently defined as “thromboinflammation”. Indeed, platelet-derived EVs interacting with monocytes can transfer glycoprotein receptors from platelets to the latter, making them susceptible to VWF released by endothelial cells [103]. Another scenario in which activated leukocytes play a crucial role in thrombosis is the formation of neutrophil extracellular traps (NETs), a network of extracellular DNA strands released by activated neutrophils to capture pathogenic microbes and facilitate their phagocytosis and killing [104]. Phosphatidylserine-decorated NETs have been described in thrombi from acute stroke patients, where they may act as structural and functional platforms for the deposition of PEVs and coagulation factors, thereby increasing thrombin and fibrin formation [105].

The induction of thrombus formation is a bidirectional process: if EVs can trigger the coagulation cascade by interacting directly with platelets and the endothelium, injured or activated endothelial cells can in turn recruit EVs by releasing VWF, which binds to GPIb and thus enhances the whole process [90]. Finally, endothelial cell- and leukocyte-derived EVs have been shown to be involved in hemostasis in its entirety, as they also modulate the final fibrinolytic phase and contribute to vascular healing. TNF-α-stimulated endothelial cells release the urokinase-type plasminogen activator receptor (uPAR), which contributes to the conversion of plasminogen to plasmin and lysis of the fibrin clot [106].

Already in healthy subjects, cell-derived EVs support the formation of thrombin to a small extent via TF-independent pathways. An interesting hypothesis is that this behavior promotes an anticoagulant scenario by allowing low levels of thrombin to mainly activate anticoagulant PC instead of triggering a full-fledged coagulation process [92]. Moreover, under physiological conditions, the prothrombotic tendency of TF is counterbalanced by TFPI, which is harbored in the same lipid rafts of EVs and attenuates the inappropriate activation of TF [107]. This homeostatic balance is obviously disturbed under pathological conditions when the balance shifts towards a prothrombotic phenotype. The mechanism underlying pathologic thrombosis is clearly different from that in the healthy state. Several studies suggest a TF-independent, low-grade activation of the coagulation cascade promoted by phosphatidylserine, which is expressed by any type of EV under physiological conditions, whereas a strong involvement of TF is restricted to pathological conditions when its expression is enhanced by proinflammatory stimuli and is limited to some specific EVs. Indeed, the highest TF activity was detected in activated monocytes, while EVs from endothelium that showed lower expression of TF and EVs from platelets did not express it at all [108,109]. Moreover, when monocytes are enriched with cholesterol, their EVs express more phosphatidylserine and TF, showing a stronger prothrombotic activity, reinforcing the link between cholesterol-rich atherosclerotic plaques and arterial thrombosis [110].

In addition to chemical stimuli, the release of EV from platelets can also be triggered by turbulent blood flow, such as that found in stenotic arteries. This confirms the close link between atherosclerosis and arterial thrombosis in a vicious cycle involving the shear stress-induced release of platelet EV, their upregulation of adhesion molecules on endothelial cells, recruitment of leukocytes and inflammation, exacerbation of atherosclerosis and stenosis and thrombosis [111,112].

### Evidence for the Involvement of EVs in Arterial and Venous Thrombosis

The intriguing link between arterial disease and thrombosis has been clearly established, and EVs may contribute, at least in part, to its explanation. Several studies have reported higher levels of circulating EVs, particularly those derived from endothelial cells, in patients with ACS such as MI or unstable angina, and an increasing concentration gradient has been described when moving from healthy controls to patients with stable angina and unstable angina to acute MI, suggesting an association with the severity of arterial injury [113]. It is noteworthy that not only the number but also the phenotype of EVs change depending on the course of the disease and possibly describes specific kinetics. For example, shortly after an acute MI, most EVs have a specific signature (CD66b+, CD62E+, CD142+).

Thrombogenic EVs have also been described in atherosclerotic plaques. They are mainly derived from monocytes, macrophages and endothelial cells, but not from platelets [114], suggesting that the former are more involved in the initial process of atherogenesis and promote the transition to an unstable atherosclerotic plaque, whereas platelet-derived EVs are more responsible for the acute event of arterial thrombus formation after interacting with local non-platelet-derived EVs within the unstable plaque. The advent of proteomics has also made it possible to define specific protein signatures capable of distinguishing EVs from patients with ACS from those of subjects with stable angina, with an upregulation of acute-phase proteins such as fibrinogen and α2-macroglobulin in the first group [115].

As far as venous thrombosis is concerned, EVs have also been shown to be markers of an altered coagulation process in this vascular territory. Higher levels of TF-expressing circulating EVs have been described, not only in patients with DVT [116] but also in individuals with a pure predisposition to VTE, such as those with factor V Leiden [117] or carriers of prothrombin gene mutations, in whom EVs may play a role in the development of VTE by increasing thrombin generation [118]. In DVT, elevated levels of TF-positive EVs have been suggested as a marker for recurrence [116]. Similarly, a prospective study in patients with primary brain tumors showed that although the concentration of EVs in those who developed PE was similar to those who did not, the pro-coagulant annexin V+ and annexin V+/CD142+ EVs were significantly higher in patients with PE than in those without PE [119], suggesting that not only the quantity but also the quality of EVs could be used for their predictive value (Figure 2).

## 7. Thrombotic Diseases and New Therapies

Antiplatelet therapy remains a fundamental component in managing patients with atherothrombotic diseases [120]. In the latest decades, several oral and intravenous antiplatelet drugs have been available for clinical use, aiming to enhance their effectiveness by reducing clinical atherothrombotic events, especially among higher-risk patients (Table 3). 

Both single and dual antiplatelet therapy (DAPT)—typically aspirin and P2Y12 inhibitor—have proven effective in reducing cardiovascular events for patients with CAD, peripheral artery disease (PAD) and cerebrovascular disease [121,122]. DAPT has demonstrated a prevalent benefit in terms of the prevention of cardiovascular events compared to single antiplatelet therapy, especially in patients with acute athero-thrombotic events (such as MI or acute stroke) and in patients undergoing percutaneous coronary or peripheral intervention. More recently, dual antithrombotic therapy, aspirin plus low-dose rivaroxaban, has proven effective in increasing survival by 30% compared with aspirin-only treatment, especially in patients with multi-vascular involvement (atherosclerosis affecting ≥2 vascular beds) [123].

However, the increased use of more potent or multiple antithrombotic agents has been accompanied by a rise in clinically significant bleeding. As a result, there is a growing interest in exploring additional strategies to achieve a greater net clinical benefit. These strategies include the development of tools to predict in individual patients the risk of bleeding and ischemic events, targeting a reduction of antiplatelet exposure for patients with low ischemic or high bleeding risk and improvements in percutaneous stent technologies to minimize the thrombotic risk [124,125]. 

### 7.1. Atrial Fibrillation and Venous Thrombosis

The tendency for blood stasis in the venous system represents a fertile ground for thrombus generation, especially when combined with other risk factors (inflammation, endothelilal disfunction, immobility, infection disease, pregnancy and genetic predisposition). These factors interact complexly, leading to thrombin activation and clot formation [126]. Thus, thrombi generated in the venous system exhibit a lower platelet count and a higher concentration of fibrin than arterial thrombi, leading to deep vein thrombosis and pulmonary embolism, collectively called VTE. A similar milieu may be generated during atrial fibrillation (AF), leading to left atrial appendage thrombosis, which is responsible for 90% of cardioembolic strokes in the case of non-valvular AF [127].

In the management of AF, the CHA₂DS₂-VASc score is employed to stratify stroke risk based on several clinical factors, including congestive heart failure, hypertension, age, diabetes, stroke history, vascular disease, and gender category, guiding the use of antithrombotic therapy [127]. 

Anticoagulant agents, including heparins and direct oral anticoagulants (DOACs), are crucial for preventing clot extension and embolization. For a decade, DOACs have offered patients a safer and readily active option. The DOACs include dabigatran, which inhibits thrombin, and rivaroxaban, apixaban and edoxaban, which inhibit factor Xa. Due to their rapid onset and offset of action, predictable dose-response, minimal interactions with other medications, no interaction with food, lower risk of major bleeding and efficacy comparable to vitamin K antagonists, DOACs are the best choice for preventing and treating VTE and for protecting from AF-related thromboembolism [128,129,130,131]. Although currently available DOACs are effective, bleeding is the most frequent side effect. 

To date, bleeding is considered an unavoidable side effect of anticoagulant therapy, secondary to inhibition of both thrombotic and hemostatic pathways. Emerging evidence suggests that contact factors may uncouple the hemostatic and thrombotic function, and factors XI and XII are potential novel therapeutic targets, due to their essential role in thrombosis and their minor role in hemostasis [132] (Table 4, Figure 3).

#### Life-Threatening Clot

In cases where there is a life-threatening clot, thrombolytic agents such as tenecteplase may be used. Tenecteplase, a tissue plasminogen activator, acts by binding to fibrin within clots and converting trapped plasminogen to plasmin, leading to the fibrin matrix’s breakdown and clot dissolution. However, it also increases the risk of bleeding complications due to its systemic effect on fibrinogen and other clotting factors, necessitating careful patient selection.

In clinical practice, this pharmacological class has limited and very well-specified indications because the high risk of bleeding associated with their administration needs to be outweighed by a much greater benefit. Thus, the indication is the presence of life-threatening clots such as acute ischemic stroke, acute myocardial infarction with ST-segment elevation if percutaneous treatment is not feasible, acute pulmonary embolism with very high risk and selected cases of acute deep vein thrombosis. In these selected populations, thrombolytic agents might enhance survival rates by 50% compared to placebo [132].

### 7.2. Therapeutic Implications of Extracellular Vesicles in Thrombosis

The well-established and strong involvement of EVs in thrombosis, in addition to their diagnostic and predictive utility, may also pave the way for future therapeutic perspectives. In other words, EVs could be used in a reverse manner by harnessing their ability to mediate intercellular communication to deliver therapeutic molecules systemically or locally, or at least inspire the development of EV-like vehicles for important drugs [133]. For example, to overcome the common off-target effect of thrombolytics, which increases the risk of hemorrhage, Pawlowski et al. have developed EV-inspired nanoparticles that encapsulate plasminogen activators to ensure their targeted delivery to the thrombus [134]. The ability of EVs to influence the phenotype of their targets through the transfer of genetic material, transcription or growth factors could also be utilized in therapies aimed at shaping the local tissue microenvironment to the benefit of the patient. Hou et al. developed a customized arterial stent consisting of a surface with electrostatically bonded nanoparticles loaded with natural factors to prevent re-stenosis and promote vessel wall remodeling. This platform not only increased the number of endothelial cells, but also improved their function in anticoagulant and anti-inflammatory directions, for example by increasing the release of NO and promoting an M2-oriented macrophage phenotype [135]. Although scientific research in the field of microvesicles is still in its infancy, it is very promising. Investigating their relationship with thrombosis and the molecular mechanisms underlying their complex interactions in the coagulation cascade and their close association with inflammation will certainly prove useful, not only in relation to cardiovascular disease but also in relation to pro-thrombotic clinical conditions in general, including cancer.

## 8. Conclusions

The balance between coagulation, fibrinolysis, the anticoagulant system and genetic factors in thrombosis is complicated and delicate. Studying thrombosis at the molecular level improves our understanding of the many processes involved. This knowledge not only sheds light on the mechanisms that trigger thrombotic events but also guides the development of groundbreaking therapeutic approaches. In the treatment of thrombotic diseases, a delicate balance must be struck between preventing in situ thrombosis and minimizing the risk of bleeding. The range of pharmacologic interventions available reflects the complexity of these clinical conditions. Ongoing research on experimental antithrombotic drugs promises to improve patient outcomes by developing new pharmacotherapies that target the specific receptors and signaling pathways involved in the thrombotic process while preserving the normal hemostatic function of platelets and the coagulation cascade.

It is crucial to tailor treatments to the individual characteristics of patients, which underlines the paramount importance of a personalized approach in the treatment of thrombotic diseases. 

## Figures and Tables

**Figure 1 ijms-25-02523-f001:**
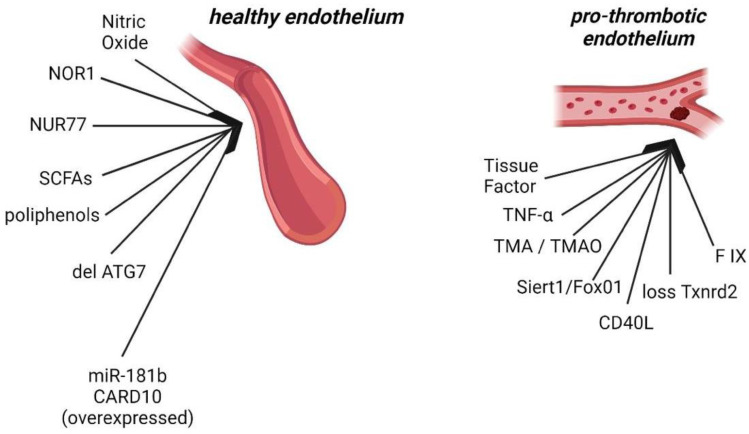
A schematic representation of some of the newest factors that act on the endothelium to make it healthy or to lead it in a thrombotic state.

**Figure 2 ijms-25-02523-f002:**
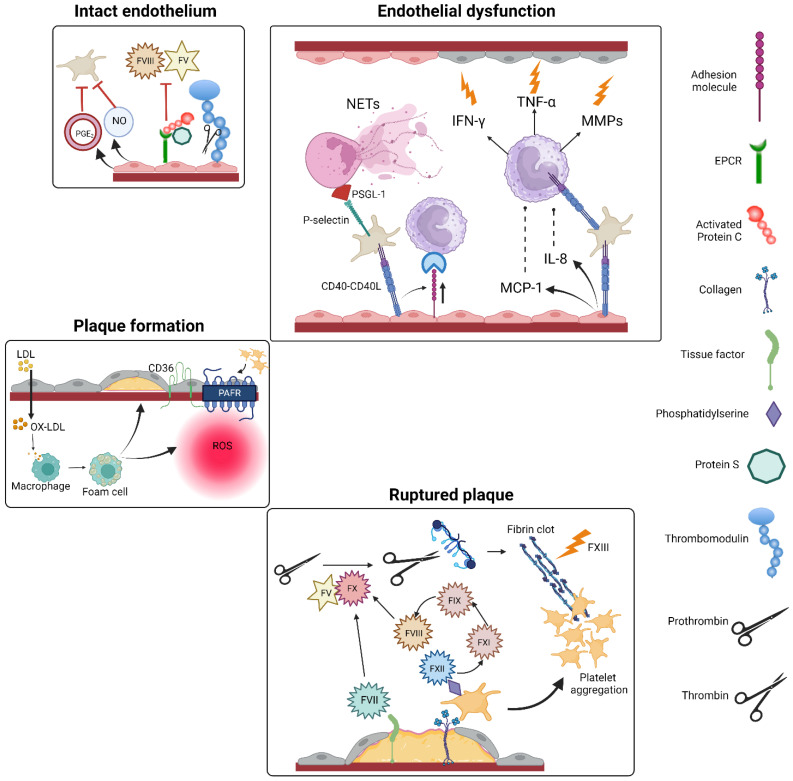
Interplay between the endothelium, the immune system and the hemostatic process during atherosclerotic plaque formation and subsequent rupture. Abbreviations: CD, cluster of differentiation; CD40-L, CD40 ligand; EPCR, endothelial protein C receptor; IFN-γ, interferon-γ; IL, interleukin; LDL, low-density lipoprotein; MCP-1, monocyte chemoattractant protein-1; MMP, matrix metalloproteinase; NET, neutrophil extracellular trap; NO, nitric oxide; OX-LDL, oxidized low-density lipoprotein; PAFR, platelet-activating factor receptor; PGE_2_, prostacyclin; PSGL-1, P-selectin glycoprotein ligand-1; ROS, reactive oxygen species; TNF-α, tumor necrosis factor-α.

**Figure 3 ijms-25-02523-f003:**
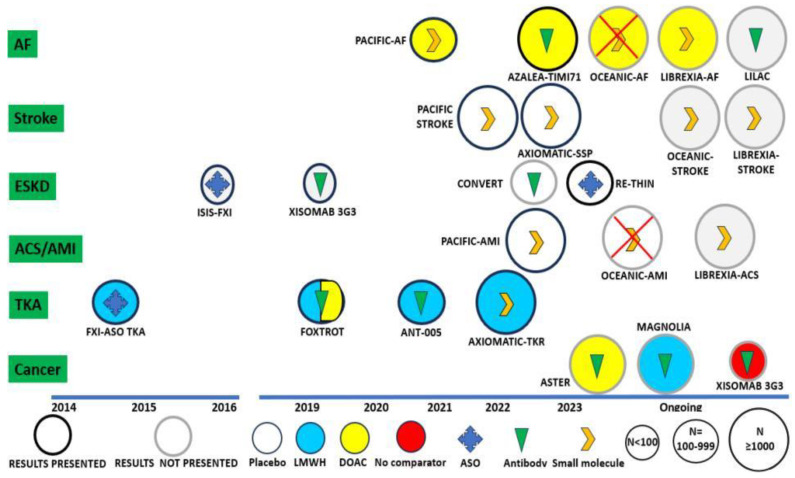
Completed and ongoing studies with factor XIa inhibitors in different clinical settings. Abbreviations: ACS, acute coronary syndrome; AF, atrial fibrillation; AMI, acute myocardial infarction; ASO, anti-sense oligonucleotide; DOAC, direct oral anticoagulant; ESKD, end-stage kidney disease; LMWH, low molecular weight heparin; TKA, total knee arthroplasty. Note: The OCEANIC-AF trial was prematurely interrupted for higher incidence of ischemic events in the asundexian arm compared to apixaban, and the OCEANIC-AMI trial is no longer ongoing.

**Table 1 ijms-25-02523-t001:** Well-established thrombosis susceptibility alleles.

Gene	Product	SNP	Position in the Gene	Risk Allele Frequency ^a^	OR/RR ^b^
ABO	Glycosyltransferases, responsible for the formation of antigens in blood type A and/or B	rs579459	Intronic (T/C *)	0.196	1.50
		rs8176749	Leu310Leu *	0.128	1.50
F2	Protrombin	rs1799963	3′UTR (G/A *)	0.0084	2.50
		rs3136516	Intronic (G/A *)	0.388	1.12
F5	Factor V	rs6025	Arg506Gln (G/A *) (FV Leiden)	0.017	3.00
		rs4524	Lys858Arg (A/G *)	0.75	1.20
F8	Factor VIII	rs114209171	Upstream (T */C)	0.80	1.11
FGG	Gamma component of fibrinogen	rs2066865	3′UTR (C/T *)	1.50	1.25
KNG1	Kininogen 1	rs7100446	Ile581Thr *	0.40	1.20
PROCR	Protein C receptor	rs6088735	Upstream (C/T)	0.23	1.11
		rs867186	Ser219Gly *	0.10	1.22
PROS1	Protein S	rs121918472	Ser501Pro *	0.002	6.57
		rs121918474	Lys196Glu *	0.009	5.00
SERPINC1	Serpin family C member 1	rs2227624	Val30Glu *	0.004	1.30
SLC44A2	Solute carrier family 44 member 2	rs2288904	Arg * 154Gln	0.76	1.28
STXBP5	Syntaxin binding Protein 5	rs1039084	Asn * 436Ser	0.55	1.11
THBD	Thrombomodulin	rs16984852	5′UTR (G/T *)	0.005	2.80
TSPAN15	Tetraspanin 15	rs78707713	Intronic (T */C)	0.90	1.42
VWF	von Willebrand factor	rs1063856	Thr789Ala *	0.40	1.15

* Risk allele. ^a^ https://gnomad.broadinstitute.org/ accessed on 12 January 2024. ^b^ Estimates of the odds ratio (OR) or relative risk (RR) associated with the risk allele.

**Table 2 ijms-25-02523-t002:** Surface markers of extracellular vesicles according to their cell of origin.

Cells of Origin	Markers
Endothelial cells	CD31 (PECAM-1), CD51 (integrin αv), CD61 (integrin β3, CD54 (ICAM-1), CD62-E (E-selectin), CD105 (endoglin), CD144 (VE-cadherin), CD146 (S-endo-1), CD106 (V-CAM1) *, CD142 (TF) *
Platelets	CD31 (PECAM-1), CD41 (integrin αIIb), CD42b (integrin Ibα), CD61 (integrin β3), CD62P (P-selectin) *, AnxA5, CD142 (TF) *, if derived from megakaryocytes: full-length filamin-A, LAMP-1
Red blood cells	CD235a (glycophorin A), CD108a (semaphorin 7A), AchE-E, AnxA5
Neutrophils	CD11b (integrin αM), CD66b (CEACAM8), CD142 (TF) *, AnxA5, CD35, MPO
Monocytes	CD11b (integrin αM), CD14, CD16 (FcγR III), CD18 (integrin β2), CD64 (FcγRI), CD142(TF) *, MPO, CD31, CD142, AnxA5
Lymphocyte	CD3, CD45

* Upon activation. Abbreviations: AchE-E, erythrocytic acetylcholine esterase; AnxA5, annexin A5; CD, cluster of differentiation; CEACAM8, carcinoembryonic antigen-related cell adhesion molecule 8; ICAM-1, intercellular adhesion molecule-1; LAMP, lysosome-associated membrane protein; MPO, myeloperoxidase; PECAM-1, platelet and endothelial cell adhesion molecule-1; TF, tissue factor; TRAP, thrombin receptor-activating protein, V-CAM (vascular cell adhesion molecule-1); VE-cadherin, vascular endothelial cadherin.

**Table 3 ijms-25-02523-t003:** Approved and investigational antiplatelet therapies for thrombotic disease.

		Name	Molecule Type	Mechanism	Half-Life	Duration of Action	Administration	Indications
**Platelet Target**	Oxygenase	**Aspirin**	Acetylsalicylic acid	Irreversible acetylation and inhibition of COX enzyme	Dose dependent	10 days	Oral, once daily	Secondary prevention of CVD; ACS ± PCI
		**Triflusal**	Acetoxy-trifluoromethylbenzoic acid	Irreversible acetylation and inhibition of COX enzyme	34 h	10 days	Oral	Prevention of ACSa dn stroke, but not EMA and FDA approved
		**ML355 ***	Small molecule	12-LOX Inhibition	2.5 h (murine)	-	Oral	-
	**ADP receptor**	**Clopidogrel**	Thienopyridine, P2Y12 antagonist	Competitive, irreversible, P2Y12 receptor blockade	6 h	5–7 days	Oral, once daily	ACS ± PCI; PCI (elective); symptomatic, high risk, CVD
		**Prasugrel**	Thienopyridine, P2Y12 antagonist	Competitive, irreversible, P2Y12 receptor blockade	7 h	7–10 days	Oral, once daily	ACS + PCI only
		**Ticagrelor**	Triazolopyrimidine, P2Y12 antagonist	Noncompetitive, reversible, P2Y12 receptor blockade	8–12 h	3–5 days	Oral, twice daily	ACS ± PCI, long term with history of ACS
		**Cangrelor**	Nonthienopyridine, ATP analogue	Noncompetitive, reversible, P2Y12 receptor blockade	3–5 min	30–60 min	Intravenous	PCI
	**Glycorotein IIb-IIIa**	**Abciximab**	Humanized mouse monoclonal ab	Fab antibody fragment that binds to IIb/IIIa receptor with high affinity and low dissociation	4 h	24–48 h	Intravenous	ACS + PCI, PCI only
		**Eptifibatide**	KGD-containing heptapeptide	Competitive, reversible IIb/IIIa receptor blockade	2.5 h	4–8 h	Intravenous	ACS ± PCI
		**Tirofiban**	Nonpeptide RGD mimetic	Competitive, reversible IIb/IIIa receptor blockade	2 h	4–8 h	Intravenous	ACS ± PCI, PCI only
	Thrombin Receptor	**Vorapaxar**	Tricyclic 3-phenylpyridine	Reversible, PAR-1 receptor blockade	5–13 days	4 weeks	Oral, once daily	History of MI or PAD
		**Atopaxar ***	Bicyclic amidine	Reversible, PAR-1 receptor blockade	22–26 h	3–5 days	Oral	-
		**BMS-986120 ***	Small molecule	Reversible, PAR-4 receptor blockade	4 h	24 h	Oral	-
	Collagen Recptor	**Revacept ***	Soluble dimeric GP VI-Fc fusion protein	-	67–137 h	7 days	Intravenous	-
	**PDE**	**Dipyridamole**	Pyrimido-pyrimidine derivative	Reversible, PDE and adenosine deaminase inhibition	10 h	-	Oral	Prevention of postoperative thromboembolic complications
		**Cilostazol**	Quinoline derivative	Reversible, PDE3 inhibition	10 h	12–16 h	Oral	Claudication in PAD

* investigational drug. Abbreviations: ACS, acute coronary syndrome; CVD, cardiovascular disease; MI, myocardial infarction; PAD, peripheral arterial disease; PCI, percutaneous coronary intervention.

**Table 4 ijms-25-02523-t004:** Investigational direct anticoagulant for thrombotic disease.

		Name	Molecule Type	Mechanism	Half-Life	Activity	Administration
**Target**	Factor XI	**IONIS-FXI_Rx_**	Antisense oligonucleotide of FXI	FXI messenger RNA inhibition	20 days	Slow and long	Sub-cutaneous (weekly)
		**Fesomersen**	Antisense oligonucleotide of FXI	FXI messenger RNA inhibition	1–122 h	Slow and long	Sub-cutaneous (weekly)
		**Osocimab**	Monoclonal antibody to FXIa	FXIa inhibition	30–40 days	Fast and long	Intravenous, sub-cutaneous (monthly)
		**Abelacimab**	Monoclonal antibody to FXI/FXIa	FXI and FXIa inhibition	25–30 days	Fast and long	Sub-cutaneous (monthly)
		**Xisomab 3G3**	Monoclonal antibody to FXI	Binds FXI and blocks activation by FXIIa	20–28 days	Fast and long	Intravenous (monthly)
		**Milvexian**	Small molecule	FXIa inhibition	11–18 h	Fast and short	Oral
		**Asundexian**	Small molecule	FXIa inhibition	16–18 h	Fast and short	Oral
	Factor XI and Factor XII	**Ixodes ricinus contact phase inhibitor**	Serine protease inhibitor	Interacts with FXIIa, FXIa, and kallikrein	-	-	-
	Factor XII	**Garadacimab** **CSL312**	Monoclonal antibody to FXIIa	FXIa inhibition	-	-	-

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
