# Peer review of "Research into New Molecular Mechanisms in Thrombotic Diseases Paves the Way for Innovative Therapeutic Approaches"

_ijms, 2024, doi:10.3390/ijms25052523_

Round 1

Reviewer 1 Report

Comments and Suggestions for Authors

This is a good review in the thrombosis and thrombo-inflammation field, it provides a comprehensive overview of the pathogenesis of thrombosis and current treatment options.

In the 7.1  "Atrial fibrillation and venous thrombosis" part, it would be advisable to refer to endothelial disfunction in AT, as well as thrombosis assessment scales such as CHADS or CHADS vasc. 

Reviewer 2 Report

Comments and Suggestions for Authors

Sara Sacchetti's review is well written and very comprehensive. It is an enormous work of synthesis covering a vast field.

minor comments:

1- some abbreviations are explained later in the manuscript, probably due to reorganisation (e.g. CHD or MI), please check all abbreviations.

2- one or two paragraphs accumulate a lot of data and would benefit from a better synthesis for an easier reading of the review.

Please see attached for additional comments.

Reviewer 3 Report

Comments and Suggestions for Authors

After carefully reading this manuscript, I definitely offer a positive feed-back. The manuscript is well written and has many qualities, among which I highlight:

the maturity of the approach to the complex problem of thrombosis

logical and coherent organization of the ideas presented, with evidence from the literature to support the statements

comprehensive and exhaustive approach to the specialized literature

anchoring of the theoretical elements with the current practice and with the possible future detections for the development of therapies.

Only a few minor recommendations should be addressed:

Line 70: “vasoactive stimuli” please consider “vasoactive mediators” or “vasoactive substances”

Line 79: It should be …tenase complex (FVIIIa-FIXa)

Line 80: It should be …prothrombinase complexe (FXa-FVa)

Line 81: It should be …factor (TF)-FVIIa activation of FX

Line 97: “endothelin” please correct “endothelin-1”

Line 451, 462: shouldn't the abbreviation VT be VTE?

FV Leiden and prothrombin G20210 favors not only venous thrombosis, but also arterial thrombosis. This aspect is worth mentioning.

Line 473-475. The assessment of VTE risk is more complex. Please consider the 2019 ESC Guidelines on Acute Pulmonary Embolism (https://doi.org/10.1093/eurheartj/ehz405)

Line 567 – 570: Please rephrase to show that the tenase complex (FVIIIa and FIXa) converts factor X to factor Xa and that prothrombinase (FVa and FXa) convers prothrombin into thrombin.

Table 3. Triflusal was developed as an antiplatelet agent. I recommend to be mentioned.  

Line 707: poly-vascular involvement Recommendation: multi-vascular involvement

Line 734-737: This section refers to fibrinolytic agents, distinct from that of antiplatelet agents and from that of anticoagulants, respectively. This pharmacological class has limited and very well specified indications because the high risk of bleeding associated with their administration must be outweighed by a much greater benefit. Such indications are acute ischemic stroke, acute myocardial infarction with ST segment elevation, acute pulmonary embolism with very high risk and selected cases of acute deep vein thrombosis. Since they are a distinct family - fibrinolytics - I recommend that they be assigned an independent paragraph. It is also indicated to specify the mechanism of the action.  

 Thank you!
